# The Effects of Auricular Acupressure on Low Back Pain, Neuropathy and Sleep in Patients with Persistent Spinal Pain Syndrome (PSPS): A Single-Blind, Randomized Placebo-Controlled Trial

**DOI:** 10.3390/ijerph20031705

**Published:** 2023-01-17

**Authors:** Yunmi Lim, Hyojung Park

**Affiliations:** 1Department of Spine Center, Samsung Medical Center, Seoul 06351, Republic of Korea; 2College of Nursing, Ewha Womans University, Seoul 03760, Republic of Korea

**Keywords:** acupressure, failed back surgery syndrome, neuropathy, pain, persistent spinal pain syndrome, sleep, spine

## Abstract

(1) Background: Various procedures were performed on patients with persistent spinal pain syndrome (PSPS), but the clinical effect and safety were insufficient. The study was to examine the effects of auricular acupressure (AA) on low back pain, neuropathy, and sleep in patients on PSPS. (2) Methods: This was a randomized, single-blind, placebo-controlled study conducted from 1 March 2022 to 31 July 2022. The participants who had at least one lumbar surgery were randomly assigned to either the experimental group (*n* = 26) or the placebo control group (*n* = 25). All participants received 6 weeks of AA intervention. To validate the effects of the intervention, pressure pain thresholds (PPT), the Visual Analogue Scale (VAS), douleur neuropathique 4 (DN4) questions, the Pittsburgh Sleep Quality Index (PSQI), and actigraphy with a Fitbit Alta were conducted. The data were analyzed with SPSS/WIN ver. 27.0, using a *t*-test and repeated-measures ANOVA. (3) Results: The findings showed that there were significant differences in pain (back VAS *p* = 0.003, leg VAS *p* = 0.002, PPT *p* = 0.008), neuropathy (DN4 *p* = 0.034), and sleep actigraphy (sleep efficiency *p* = 0.038, number of awake *p* = 0.001, deep sleep stage *p* = 0.017). (4) Conclusions: We conclude that AA is an effective, safe, cost-effective, non-invasive nursing intervention that can improve pain, neuropathy, and sleep in patients on PSPS.

## 1. Introduction

Many definitions and revisions of the term failed back surgery syndrome (FBSS) have been proposed, and recently, a more appropriate and clinically informative replacement term, persistent spinal pain syndrome (PSPS), was proposed [1]. PSPS indicates persistent, new, or recurrent low back and/or lower extremity pain following one or more spine surgeries [1]. PSPS is a condition that may cause radicular pain with persistent pain even after spinal surgery, and it occurs in about 10–40% of surgical patients [2]. The International Association for the Study of Pain (IASP) and the World Health Organization (WHO) have newly added postsurgical pain to the International Classification of Diseases, 11th Revision (ICD-11) [3]. In general, chronic pain lasts more than 6 months, but chronic pain after spinal surgery is defined as pain lasting more than 3 months. This takes into account the rapid appearance of neuropathic pain after spinal surgery, indicating that rapid pain management after surgery is important. PSPS is most common in lumbar spine surgery [1]. Its symptoms include surgical pain around the surgical site and radiating pain down to the lower extremities, and it may progress into postsurgical pain syndrome if there was a neurological deficit, peripheral neuropathy, cauda equine syndrome, etc., before surgery [3]. Neuropathy presents also at rest, characterized by more severe pain at night, and more than 50% of people who have undergone spinal surgery complain of sleep disorders after spinal surgery along with neuropathy [4,5]. Neuropathy was used as a douleur neuropathique 4 (DN4) question. Patients with postsurgical pain syndrome experience neurological symptoms along with pain, and if such symptoms persist for a long time, it may cause mental and psychological disorders such as depression, anxiety, decreased quality of life, etc., requiring active management [5,6].

Recently, patients with PSPS are interested in complementary and alternative approaches whose effect and safety have been proven [7]. Among complementary and alternative therapies, AA has been proven to be effective for a variety of pains, such as postoperative pain after surgery [8], cervicalgia [9], lumbago [10], etc., and are frequently utilized. AA is a representative complementary and alternative medicine used for various diseases based on acupuncture [11]. Auricular acupressure (AA) is an economical, simple, and non-invasive intervention that does not have the risk of cross-infection or side effects; thus, it is considered a nursing intervention that can be performed independently by nurses instead of acupuncture for patients who have undergone spinal surgery [12]. In addition to pain, AA has been proven effective in sleep disorders [13] and neuropathy [14].

Although there are relatively many studies that applied AA to those with back pain and proved its effects, spinal surgery was excluded from the criteria for the subject selection, and there was no mention made as to whether or not a surgery was performed [15,16]. Since only some of the surgical patients were included [10], it was difficult to confirm its effect on only those who had undergone spinal surgery.

AA can reduce subjective pain [10]. The Visual Analogue Scale (VAS) is generalized to measure subjective pain severity [16]. In addition, AA has a pain-control effect by increasing the pressure pain threshold (PPT), and the study of measuring PPT at a certain area using a pressure algometer can be said to be a scientific and valid approach [10]. Sleep assessment can increase the validity of studies by utilizing objective indicators that can quantify sleep along with subjective indicators such as Pittsburgh Sleep Quality Index (PSQI). Recently, research using sleep actigraphy has been actively conducted.

In this study, AA was applied for 6 weeks to patients with PSPS to determine its effect on pain, neuropathy, and sleep. The subjects were randomly assigned to the placebo-controlled group and the experimental group, and the effect of AA was examined through an objective and scientific approach by using a pressure algometer, which is a subjective and objective indicator, and a Fitbit tracker, a device that measures sleep–wake activity. 

The purpose of this study is to determine the effect of AA on pain, neuropathy, and sleep by applying it to patients with PSPS and to establish the basis for independent nursing interventions.

## 2. Materials and Methods

### 2.1. Study Design and Participants

This is a randomized, single-blinded, placebo-controlled pre-test and post-test design experiment. The study was conducted from 1 March 2022 to 31 July 2022. The study subjects were spine surgery patients who wished to voluntarily participate in the study through a recruitment announcement at the spine center of a tertiary general hospital located in S City. After the researcher provided an explanation of the study to the participants and obtained written consent directly from them, only those who satisfied both the selection and exclusion criteria were selected as the subjects of the study. 

The selection criteria for subjects are: (a) those who are over the age of 18 and have undergone one or more spinal surgeries; (b) those who have VAS on back and leg pain over 3 points and persistent pain for more than 3 months; (c) those who have DN4 questions more than 4; (d) those who have a PSQI score higher than 5 points; and (e) those who can read and respond to the questionnaires. The exclusion criteria are: (a) those who take sleeping pills and depression medications; (b) those who have been diagnosed with diabetes; (c) cancer patients receiving chemotherapy; (d) those who are currently undergoing or have experience with other complementary and alternative therapies; or (e) those who have ear inflammation or are allergic to band-aids and other bandages.

The sample size was calculated by using G power 3.1.9.7. Based on a prior study that used the PSQI [17], the effect size was set at 0.73, and through the one-sided test with a significance level (α) of 0.05 and a power of 0.80, the number of patients needed to examine the average group difference was calculated to be 25 people for the experimental group and the control group, respectively. Since the study period was limited to six weeks, a drop-out rate of 20% was considered. Thus, a total of 60 people were needed for this study, 30 subjects for each of the groups. So, 60 people were recruited as the study subjects; however, a total of 56 people were selected because 4 people did not meet the selection criteria with less than 4 points in neuropathy. Fifty-six people were randomly assigned to the experimental group and control group by using the random-number table generated from the Random Allocation Software (Version 1.0). The subjects were treated single-blind to prevent them from knowing which group they belonged to. Two people out of the 28 people in the experimental group dropped out as they received acupuncture; 2 people out of the 28 people in the control group changed their medications; and 1 person dropped out due to the long distance traveled to participate in the study. Therefore, the final number of people for the data analysis was 26 in the experimental group and 25 in the control group (Figure 1).

### 2.2. Intervention

In order to collect the data for this study, the researcher, who works at the spine center as a nurse, has 20 years of clinical experience at a tertiary general hospital, is fully knowledgeable about the physiology of human anatomy and major body mechanisms, particularly on the spine, and has a high understanding of patients who are expecting and have undergone a spinal surgery, completed a specialized course in AA conducted by the Korean Nurses Association of Complementary and Alternative Therapy and obtained an expert certificate. In the research procedure, the researcher performed hand hygiene before AA and wiped the foreign substances in one ear of the patients with a disinfectant cotton pad moistened with sterile saline solution. Then, any abnormal findings were observed in the ear. For the AA, *Sinapsis alba* seeds were attached to 3M paper tape. Before the intervention, the reflex points were disinfected using a 75% alcohol cotton pad, and then the *Sinapsis alba* seeds were applied [12]. *S. alba* seeds are plant seeds with a warm nature, smooth surface, and uniform size and hardness, so they are mainly used in AA [9,12]. In a systematic literature review of chronic back pain, the AA application period was recommended from 4 weeks to 12 weeks [16], and a study of AA application for 6 weeks showed a significant reduction in pain from 5 weeks onwards. Therefore, an AA of 6 weeks was determined in this study [10]. The AA intervention was provided for 6 weeks with 5 days of seed attachment and 2 days of rest as 1 cycle. The reflex points of the experimental group were shenmen, subcortex, sympathetic nerves, liver, and lumbar vertebrae, whereas using helix 1~5 points, which was the zone not corresponding to pain, neuropathy, and sleep and away from the reflex points of the experimental group, for the control group (Figure 2) [12,14]. The suitability of these points was consulted with two experts before the final selection. In order to confirm the accuracy of the AA, the reflex points and intervention procedure to be used in this study were examined by an expert from the Korean Nurses Association of Complementary and Alternative Therapy. Before conducting the study, permission was obtained from the director of the institution and two spine center/orthopedic specialists.

For the 5 days of attaching the AA patches, the study subjects were instructed to press the pressure-applied areas four times a day for 2 min using their thumb, index, or middle finger as hard as to feel tingling [16]. The same procedure and method were applied to the placebo-controlled group.

### 2.3. Measures

Back VAS, leg VAS, and lumbar PPT were measured weekly to measure the subjects’ pain. The PPT was measured using a digital algometer (Wagner Instrument FPX25, Greenwich, CT, USA). PPT was measured in a treatment room with a bed. For the PPT measurement, the subjects were guided to lie face down in a comfortable position on the bed after listening to the measurement method and then asked to point out the point where the back pain is most severe with one finger. After checking the subjects’ most painful point by pressing the area with a finger, the area was pressed in the vertical direction at a constant speed and pressure using the pressure algometer. Then, the subjects were asked to signal the researcher at the moment when they felt pain to measure the value applied at that moment (kg/cm^2^). In order to reduce the error of the measurer, the average value was recorded by repeatedly measuring three times at 1 min intervals. Neuropathy was identified by the DN4 questions [18] developed by Bouhassira et al. (2005). Sleep was measured with the PSQI [19] and the Fitbit Alta HRTM (FitBit^®^ Inc., San Francisco, CA, USA), an actigraphy meter. The PSQI scores range from 0 to 21 points, and the higher the total score, the lower the sleep quality. A score of more than 5 points indicates a sleep problem. Actigraphic is a value that is analyzed by detecting heart rate, wrist motion monitoring, and heart-related factors. In this study, the values that were measured by synchronizing the Fitbit app and Fitbit Alta HR device on an Android phone were collected and the sleep items were downloaded from the official website (www.fitbit.com, accessed on 1 March 2022) as Excel files for further data collection.

### 2.4. Ethical Considerations

Before starting the study, approval from the S tertiary general hospital (SMC 2021-10-104-002) was obtained to examine the ethical validity of the research protocol. The subjects were explained the purpose of the study, research methodology, randomization, testing and procedures, compliance details, possible side effects, benefits, participation withdrawal, discontinuation, and privacy in an easy-to-understand manner, and then written consent was obtained from them. The researcher explained that research data are confidential and would be used for research purposes only, and that participation in the study could be withdrawn at any time during the study. They were also told that there was no disadvantage whatsoever. During the study participation period, the subjects were asked not to receive any complementary and alternative therapies other than AA and were educated to consult with the researcher if they faced a situation where they needed to receive other treatment during the study participation. For such purposes, the researcher’s contact information was provided to the subjects. Allergic reactions such as itching may sometimes occur due to AA; therefore, the subjects were also asked to contact the researcher immediately if they experience an allergic reaction below the minimum risk or discomfort. The researcher explained that itchiness around the seed-applied area would subside after removing the patches, and the AA might continue by replacing the paper tape only when the subjects wanted to keep participating in the study.

If the homogeneity is examined by assessing the subjects’ medical records on the medication and their symptoms are significantly reduced only in the experimental group throughout the study period without any change in their medication, such a change is considered to be the effect of AA. It is because the reflex points that had nothing to do with the variables used in the study were applied to the control group.

All participants were paid for transportation expenses as compensation, and the same AA, that was applied to the experimental group, was provided to the control group as well after the study. Due to COVID-19 (coronavirus disease 2019), the researcher informed the subjects of a possible suspension of the study according to quarantine rules and educated them to follow the quarantine rules to prevent infection. Both the researcher and the subjects thoroughly observed the personal quarantine rules such as hand hygiene and mask wearing during the study.

### 2.5. Analysis

The collected data were analyzed using the SPSS WIN 27.0 The homogeneity of the experimental and control groups was analyzed using the chi-square test, Fisher’s exact test, independent *t*-test, and Mann–Whitney U test. The normality of the dependent variables was analyzed using the Shapiro–Wilk normality test. For assessment of the study hypotheses, the differences between the experimental and control groups were analyzed with an independent *t*-test or the Mann–Whitney U test, while the intra-group differences were examined with the paired *t*-test or the Wilcoxon signed-rank test. Changes over time were analyzed using repeated-measures ANOVA or generalized estimating equations.

## 3. Results

### 3.1. Characteristics of the Participants

The total number of participants in this study was 51: 26 in the experimental group and 25 in the control group. The mean age of the group was 65.81 ± 4.80 years in the experimental group and 66.32 ± 6.58 years in the control group, and the postsurgical follow-up period was 39.00 ± 26.44 months in the experimental group and 36.04 ± 23.54 months in the control group (Table 1). 

The back VAS measured before starting the experimental group was 6.38 ± 2.08 points in the experimental group and 6.28 ± 1.67 points in the control group. The leg VAS measured before starting the intervention was 6.54 ± 2.02 points in the experimental group and 6.44 ± 1.60 points in the control group, while the PPT was 5.90 ± 1.56 points in the experimental group and 6.14 ± 0.97 points in the control group. DN4 scored 4.04 ± 1.28 points in the experimental group and 4.44 ± 0.91 points in the control group, while PSQI was 12.69 ± 3.29 points in the experimental group and 12.88 ± 3.95 points in the control group (Table 2). Therefore, the homogeneity of the two groups was confirmed.

### 3.2. Effects of AA Intervention on VAS and PPT

VAS and PPT were repeatedly measured weekly before and after the AA intervention, and the results were as follows: Table 3. The back VAS (Wald χ^2^ = 20.055, *p* = 0.003), leg VAS (Wald χ^2^ = 20.777, *p* = 0.002), and PPT (F = 5.213, *p* = 0.001) showed statistically significant differences in the interaction between group and time. When analyzing the difference in the amount of change between the two groups, there was a statistically significant difference in the back VAS after 5 weeks of the intervention (Z = −2.650, *p* = 0.008), leg VAS after 5 weeks of the intervention (Z = −2.080, *p* = 0.038), and PPT after 3 weeks of the intervention (*t* = 2.109, *p* = 0.041) (Figure 3).

### 3.3. Effects of AA Intervention on DN4

The DN4 was measured twice before and after the AA intervention, and the results were as follows: Table 4. There was a statistically significant decrease between the two groups, showing a 0.84 ± 1.28-point decrease in the experimental group from 4.04 ± 1.28 points before the intervention to 3.19 ± 1.09 points after 6 weeks of the intervention (Z = −2.121, *p* = 0.034).

### 3.4. Effects of AA Intervention on PSQI and Actigraphy

The PSQI and actigraphy were measured twice before and after the AA intervention, and the results were as follows: Table 5. PSQI, a subjective indicator of sleep, showed no difference between groups after 6 weeks of the intervention (*t* = −0.752, *p* = 0.456). There were significant differences in sleep actigraphy, which is an objective indicator of sleep, when it came to sleep efficiency (*t* = 2.135, *p* = 0.038), number of awakenings (*t* = −3.627, *p* = 0.001), and deep sleep (*t* = 2.464, *p* = 0.017).

## 4. Discussion

In this study, the effect of AA on back and leg pain, neuropathy, and sleep was examined and confirmed after it was applied to patients with PSPS for 6 weeks using subjective and objective indicators. 

Back and leg VAS were reduced after 5 weeks of AA, while PPT decreased after 3 weeks of the intervention. Such a result is similar to the outcome of the studies that reported consistent pain reduction over time in the experimental group by applying AA to the elderly with arthritis for 8 weeks [20] and the elderly with chronic low-back pain for 6 weeks [10]. In a study that was conducted for the elderly with back pain [10], the PPT became significantly increased from the fifth week of the AA. Moreover, a study conducted for the elderly with arthritis reported significantly increased PPT in the back and knee-joint pain from the third week of the AA and increased PPT in their shoulder pain from the fourth week of the AA [20]. Such findings are similar to the results of this study, which show that PPT tends to positively increase over time. Looking at seven prior randomized controlled trials on low-back pain, these suggested 4–12 weeks of AA application, which is a somewhat long period to treat chronic low-back pain [16]. This study is significant in that it suggested a relatively accurate time for AA to have an effect through repeated weekly measurements of pain. 

Neuropathy decreased significantly in both groups after the intervention, showing significant differences between the two groups. These results are similar to the findings of prior studies that reported decreased neuropathy after applying AA to those with cervical spine and spinal cord injuries [21,22]. Postsurgical neuropathy is caused by a pulled spinal nerve due to postsurgical adhesion, ischemia, and obstruction of cerebrospinal fluid flow that nourishes the surrounding nerves [23]. Regarding these causes, it can be said that the reflex point shenmen acts on the central nervous system to control the sensitization of neuropathy, while the reflex points subcortex and sympathetic nerves stimulate the cerebral cortex and the autonomic nervous system to relieve neuropathic pain [11].

The PSQI score was similar to that of spinal stenosis surgery patients [24] and higher than that of spinal cord stimulation patients, suggesting that the subjects of this study had severe sleep disorders [25]. Neuropathy was reported in 48% of patients with chronic pain, and sleep disturbance was more severe in patients with neuropathy [26]. Neuropathy was characterized by worsening at night, and one of the frequent complaints of the subjects during the study period was ‘frequent electrical convulsions at night’, and it was found that there was a sleep disorder due to neuropathy. It was thought that further studies on sleep and neuropathy in this study subject would be necessary. 

There was no difference between the two groups in the PSQI, which is a subjective indicator. These results differed from the studies in which AA significantly reduced PSQI scores [13,20]. This is because the subjects in this study were taking GABA-based drugs such as gabapentin and pregabalin, so the GABA level was relatively high, resulting in the fact that the effect of AA was not seen. Importantly, 52–57% of the study subjects were taking painkillers, anti-inflammatory drugs, and neuropathy drugs. In particular, GABA, a neuropathic drug, is a neurotransmitter that regulates sleep, and it is reported that GABA is about 30% lower in people with sleep disorders [27]. AA is described through a mechanism that improves sleep by increasing GABA levels to activate parasympathetic nerves [11]. In further studies that apply AA for sleep, it is necessary to consider controlling GABA-based drugs along with a close examination of the subjects’ medications. The sleep actigraphy, an objective indicator of sleep, was measured using a Fitbit tracker. The results obtained from the subdomain of the sleep actigraphy showed significant differences in deep sleep, number of awakenings, and sleep efficiency between the two groups. These results are similar to studies in which AA applied to the elderly was effective on sleep efficiency [28], on the number of awakenings, sleep efficiency [20], deep sleep, the number of awakenings, total sleep time, and sleep efficiency [29]. Through this study, it was confirmed that AA significantly increased the duration of deep sleep by 11.50 min. Looking at prior studies, 6 weeks of AA increased deep sleep in the elderly by 23.96 min [30] and 8 weeks of AA significantly increased deep sleep in dialysis patients by 9 min [29], showing similar results as the findings of this study. The deep sleep phase is characterized by slowed heartbeats and a state of complete relaxation, which makes awakening difficult. It is the stage that contributes to the recovery and enhancement of the body through tissue restoration and the strengthening of the immune system. It is a stage that. Therefore, AA is said to contribute to physical recovery and health promotion by increasing the subjects’ deep sleep.

The reflex points used in this study were the shenmen, subcortex, sympathetic nerves, liver, and lumbar vertebrae, and a *Sinapsis alba* seed was applied to these five places to confirm the effects of pain, neuropathy, and sleep improvement. According to a systematic review on acupuncture [31], the reflex points for back pain were consistent with those of this study, and other prior studies reported that the reflex points of the shenmen, subcortex, and sympathetic nerves were effective in improving neuropathy and sleep [21,22], supporting the findings of this study.

In a study on pain after spinal surgery, it failed to prove the effect of AA after applying it to the patients undergoing spinal surgery for three days after the surgery; however, AA was proven to be effective in a study that reported a significant decrease in the amount of painkiller use [32] and two other studies showing significant pain reduction [33,34]. Although there are studies about the effect of AA on pain reduction immediately after surgery, studies on patients with pain syndrome who have undergone spinal surgery before a certain period of time are insufficient. However, this study is the only one that has examined such an area. Due to the recent interest in patients who have undergone spinal surgery, research protocols for the application of acupuncture and auricula acupressure have been published [35,36], and therefore, it is significant that this study proactively applied AA for 6 weeks to the patients with pain syndrome who had an elapsed surgical period after spinal surgery to confirm the effectiveness and safety of AA.

During the 6 weeks of AA, no one dropped out due to side effects of AA. It has been confirmed that AA is a safe and effective intervention to apply as a nursing intervention since it is non-invasive and has no risk of cross-infection, unlike acupuncture. In particular, it is significant that AA is an effective intervention to be performed by nurses for patients with PSPS who are limited to receiving an interventional treatment or who have a hard time controlling pain due to fear of receiving treatment.

As a result, AA is an effective nursing intervention for pain, neuropathy, and sleep in patients with PSPS, and it can be said to be a nursing intervention suitable for improving the quality of life of the patients.

Despite the several positive findings of AA, the study limitations need to be considered. First, mechanoreceptors can be activated only by skin stimulation, so care has to be taken in interpreting the results. Second, this study was single-blind, allowing for the risk of a placebo effect. In the future, a long-term repeat study with a more stringent PSPS is suggested.

## 5. Conclusions

In this study, AA was applied for 6 weeks to patients with PSPS to determine its effect on pain, neuropathy, and sleep. The study has shown that AA is effective for back VAS, leg VAS, PPT, DN4, and actigraphy, except for the PSQI. It is significant that this study measured pain and sleep using a combination of subjective and objective indicators. It is expected that the above study results can be used more effectively through further research on patients with PSPS using AA.

## Figures and Tables

**Figure 1 ijerph-20-01705-f001:**
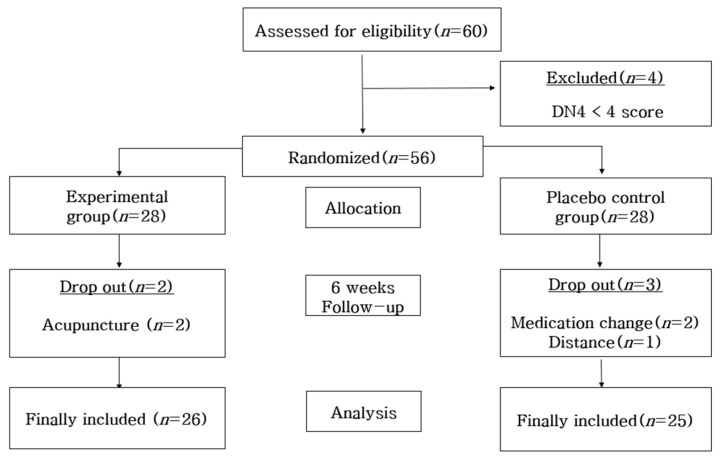
Participant selection flow.

**Figure 2 ijerph-20-01705-f002:**
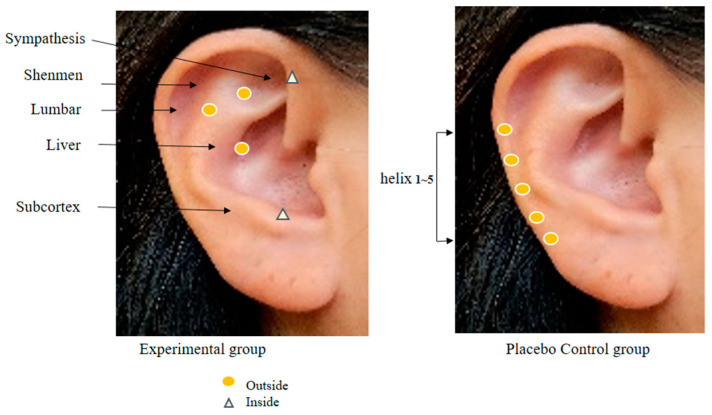
Auricular acupoints used in two groups.

**Figure 3 ijerph-20-01705-f003:**
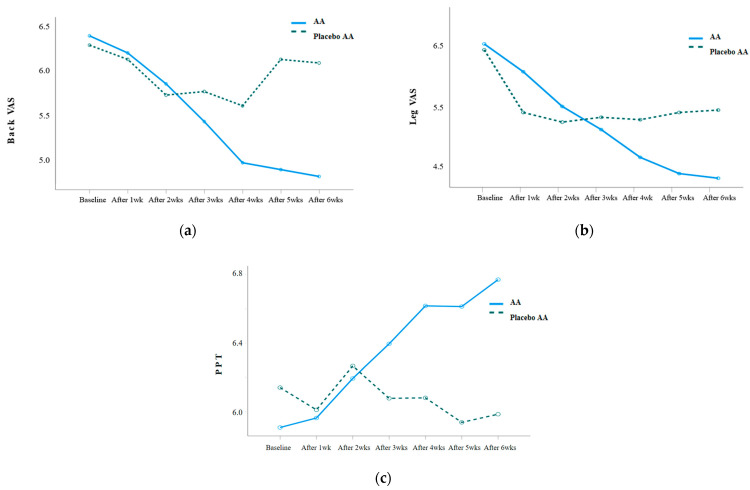
(**a**) Changes in back VAS in the experimental and placebo control groups; (**b**) changes in leg VAS in the experimental and placebo control groups; (**c**) changes in PPT in the experimental and placebo control groups.

**Table 1 ijerph-20-01705-t001:** Homogeneity of characteristics of participants (*n* = 51).

Characteristics	Categories	Experimental Group (*n* = 26)	Control Group (*n* = 25)	*t* or χ^2^	*p*
		Mean ± SD or *n* (%)			
Gender	Male	4 (15.4)	4 (16.0)		>0.999 ^†^
	Female	22 (84.6)	21 (84.0)		
Age (year)		65.81 ± 4.80	66.32 ± 6.58	−0.319	0.751
BMI (kg/m^2^)		25.66 ± 3.74	25.33 ± 3.47	0.325	0.747
Drinking	Yes	2 (7.7)	6 (27.0)		0.140 ^†^
	No	24 (92.3)	19 (76.0)		
Smoking	Yes	3 (11.5)	2 (8.0)		>0.999 ^†^
	No	23 (88.5)	23 (92.0)		
Post operation	Duration (month)	39.00 ± 26.44	36.04 ± 23.54	0.422	0.675
	≤12	3 (11.5)	2 (8.0)	1.032	0.794 ^†^
	13~36	11 (42.3)	14 (56.0)		
	37–60	6 (23.1)	4 (16.0)		
	≥61	6 (23.1)	5 (20.0)		
Medication					
COX-2 inhibitor	Yes	13 (50.0)	9 (36.0)	1.018	0.313
	No	13 (50.0)	16 (64.0)		
Analgesics	Yes	17 (65.4)	15 (60.0)	0.158	0.691
	No	9 (34.6)	10 (40.0)		
Anticonvulsants	Yes	15 (57.7)	13 (52.0)	0.167	0.683
	No	11 (42.3)	12 (48.0)		
Operation type	Fusion	16 (61.5)	14 (56.0)	0.161	0.688
	Decompression	10 (38.5)	11 (44.0)		
Surgical level ^‡^	Total count	1.69 ± 0.73	1.64 ± 0.63	0.271	0.788
	L1-2 level	1 (3.8)	0 (0.0)		
	L2-3 level	2 (7.7)	4 (16.0)		
	L3-4 level	11 (42.3)	6 (27.0)		
	L4-5 level	20 (76.9)	17 (68.0)		
	L5-S1 level	10 (38.5)	14 (56.0)		

Note: BMI = body mass index; COX-2 = cyclooxygenase-2; L = lumbar; S = sacral; SD = standard deviation; ^†^ Fisher’s exact test; ^‡^ multiple responses.

**Table 2 ijerph-20-01705-t002:** Homogeneity of dependent variables between groups (*n* = 51).

Variables	Experimental Group (*n* = 26)	Control Group (*n* = 25)	*t* or Z	*p*
	Mean ± SD/Median (IQR)			
Pain				
Back VAS	6.38 ± 2.08 6.50 (3.25)	6.28 ± 1.67 6.00 (3.00)	0.010 ^†^	0.992
Leg VAS	6.54 ± 2.02 7.00 (3.00)	6.44 ± 1.60 7.00 (3.00)	0.483 ^†^	0.629
Back PPT (kg/cm^2^)	5.90 ± 1.56	6.14 ± 0.97	−0.642	0.524
Neuropathy				
DN4	4.04 ± 1.28 4.00 (1.00)	4.44 ± 0.91 5.00 (1.00)	−1.482 ^†^	0.138
Sleep				
Self-reported				
PSQI	12.69 ± 3.29	12.88 ± 3.95	−0.185	0.854
Actigraphy				
TST (min)	328.31 ± 60.91	332.76 ± 72.52	−0.238	0.813
SE (%)	84.38 ± 4.03	83.97 ± 3.13	0.408	0.685
SL (min)	6.04 ± 3.75	7.28 ± 5.20	−0.980	0.332
TOA (min)	61.08 ± 20.39	63.60 ± 18.38	−0.463	0.645
NOA	24.27 ± 7.37	27.12 ± 5.77	−1.533	0.132
REM sleep (min)	61.12 ± 19.81	66.40 ± 23.34	−0.873	0.387
Shallow sleep (min)	214.31 ± 49.17	217.36 ± 77.92	−0.167	0.869
Deep sleep (min)	52.88 ± 18.59	49.00 ± 13.75	0.846	0.402

Note: DN4 = douleur neuropathique 4; IQR = inter-quartile range; NOA = number of awakenings; PPT = pressure pain threshold; PSQI = Pittsburgh Sleep Quality Index; REM = rapid eye movement; SD = standard deviation; SE = sleep efficiency; SL = sleep latency; TOA = time of awakenings; TST = total sleep time; VAS = Visual Analogue Scale; ^†^ Mann–Whitney U test.

**Table 3 ijerph-20-01705-t003:** Comparison of pain-related dependent variables between groups (*n* = 51).

Back VAS					Leg VAS				Lumbar PPT (kg/cm^2^)			
Day	Experimental Group (*n* = 26)	Control Group (*n* = 25)	Z ^†^	*p*	Experimental Group (*n* = 26)	Control Group (*n* = 25)	Z ^†^	*p*	Experimental Group (*n* = 26)	Control Group (*n* = 25)	*t*	*p*
	Mean ± SD/ Estimated Mean ± SE				Mean ± SD/ Estimated Mean ± SE				Mean ± SD			
Pre	6.38 ± 2.08 6.38 ± 0.40	6.28 ± 1.67 6.28 ± 0.32			6.54 ± 2.04 6.54 ± 0.38	6.44 ± 1.60 6.44 ± 0.31			5.90 ± 1.56	6.14 ± 0.97		
1	6.19 ± 2.17 6.19 ± 0.41	6.12 ± 1.45 6.12 ± 0.28	−0.437	0.662	6.08 ± 2.01 6.08 ± 0.38	5.40 ± 1.91 5.40 ± 0.37	−1.092	0.275	5.96 ± 1.41	6.01 ± 1.49	0.740	0.463
2	5.85 ± 1.91 5.85 ± 0.36	5.72 ± 1.51 5.72 ± 0.29	−0.721	0.471	5.50 ± 2.14 5.50 ± 0.41	5.24 ± 2.04 5.24 ± 0.40	−0.500	0.617	6.19 ± 1.47	6.26 ± 1.21	0.674	0.504
3	5.42 ± 1.67 5.42 ± 0.32	5.76 ± 1.61 5.76 ± 0.31	−0.851	0.395	5.12 ± 1.98 5.12 ± 0.38	5.32 ± 2.03 5.32 ± 0.39	−0.430	0.667	6.39 ± 1.63	6.07 ± 1.11	2.109	0.041
4	4.96 ± 0.99 4.96 ± 0.19	5.60 ± 1.70 5.60 ± 0.33	−1.224	0.221	4.65 ± 1.81 4.65 ± 0.34	5.28 ± 2.20 5.28 ± 0.43	−1.204	0.228	6.61 ± 1.38	6.08 ± 1.26	2.703	0.009
5	4.88 ± 0.95 4.88 ± 0.18	6.12 ± 1.42 6.12 ± 0.27	−2.650	0.008	4.38 ± 1.65 4.38 ± 0.31	5.40 ± 2.06 5.40 ± 0.40	−2.080	0.038	6.61 ± 1.22	5.93 ± 1.33	3.233	0.002
6	4.81 ± 1.38 4.81 ± 0.27	6.08 ± 1.15 6.08 ± 0.22	−2.691	0.007	4.31 ± 1.80 4.31 ± 0.47	5.44 ± 1.93 5.44 ± 0.38	−2.318	0.020	6.77 ± 1.20	5.98 ± 1.30	4.207	<0.001
	Source	Wald χ^2^	*p*		Source	Wald χ^2^	*p*		Source	F	*p*	
	Group	1.785 ^†^	0.182		Group	0.350 ^†^	0.554		Group	0.686	0.412	
	Time	20.191 ^†^	0.003		Time	44.758 ^†^	<0.001		Time	2.936	0.024	
	G × T	20.055 ^†^	0.003		G × T	20.777 ^†^	0.002		G × T	5.213	0.001	

Note: G × T = group × time; PPT = pressure pain threshold; SE = standard error; SD = standard deviation; VAS = Visual Analogue Scale; ^†^ Wald χ^2^ by generalized estimating equation.

**Table 4 ijerph-20-01705-t004:** Comparison of neuropathy-related dependent variables between groups (*n* = 51).

Variable	Group	Pre-Test	Post-Test	Mean Difference	Within Group		Between Group	
		Mean ± SD Median (IQR)	Mean ± SD Median (IQR)	Mean ± SD Median (IQR)	Z ^†^	*p*	Z ^‡^	*p*
DN4	Experimental Group (*n* = 26)	4.04 ± 1.28 4.00 (1.00)	3.19 ± 1.09 3.00 (2.00)	−0.84 ± 1.28 −1.00 (−1.25)	−2.855	0.004	−2.121	0.034
	Control Group (*n* = 25)	4.44 ± 0.91 5.00 (1.00)	3.84 ± 0.85 4.00 (1.5)	−0.60 ± 0.76 0.00 (−1.00)	−3.035	0.002		

Note: DN4 = douleur neuropathique 4; IQR = inter-quartile range; SD = standard deviation; ^†^ Wilcoxon signed-rank test; ^‡^ Mann–Whitney U test.

**Table 5 ijerph-20-01705-t005:** Comparison of sleep-related dependent variables between groups (*n* = 51).

Variable	Group	Pre-Test	Post-Test	Mean Difference	Within Group		Between Group	
		Mean ± SD Median (IQR)	Mean ± SD Median (IQR)	Mean ± SD Median (IQR)	*t*	*p*	*t*	*p*
Self-reported								
PSQI	Exp (*n* = 26)	12.69 ± 3.29	11.08 ± 3.75	−1.61 ± 3.91	−2.107	0.045	−0.752	0.456
	Con (*n* = 25)	12.88 ± 3.95	11.84 ± 3.48	−1.04 ± 3.96	−1.312	0.202		
Actigraphy								
TST (min)	Exp (*n* = 26)	328.31 ± 60.91	372.69 ± 69.54	44.38 ± 93.58	2.418	0.023	1.352	0.183
	Con (*n* = 25)	332.76 ± 72.52	348.44 ± 57.79	15.68 ± 66.41	1.180	0.249		
SE (%)	Exp (*n* = 26)	83.38 ± 4.03	87.30 ± 3.74	2.91 ± 4.63	3.206	0.004	2.135	0.038
	Con (*n* = 25)	83.97 ± 3.13	85.29 ± 2.91	1.31 ± 4.04	1.632	0.116		
SL (min)	Exp (*n* = 26)	6.04 ± 3.75	5.38 ± 3.87	−0.65 ± 5.49	−0.607	0.549	−0.667	0.508
	Con (*n* = 25)	7.28 ± 5.20	6.08 ± 3.54	−1.20 ± 6.24	−0.961	0.346		
TOA (min)	Exp (*n* = 26)	61.08 ± 20.39	53.73 ± 17.83	−7.34 ± 22.75	−1.646	0.112	−1.368	0.177
	Con (*n* = 25)	63.60 ± 18.38	60.44 ± 17.15	−3.16 ± 18.77	−0.842	0.408		
NOA	Exp (*n* = 26)	24.27 ± 7.37	21.31 ± 6.16	−2.96 ± 6.81	−2.216	0.036	−3.627	0.001
	Con (*n* = 25)	27.12 ± 5.77	28.36 ± 7.66	1.24 ± 6.99	0.886	0.384		
REM	Exp (*n* = 26)	61.12 ± 19.81	64.62 ± 25.69	3.50 ± 34.37	0.519	0.608	−0.344	0.732
Sleep (min)	Con (*n* = 25)	66.40 ± 23.34	66.88 ± 20.97	0.48 ± 29.28	0.082	0.935		
Shallow	Exp (*n* = 26)	214.31 ± 49.17	243.69 ± 61.08	29.38 ± 75.56	1.983	0.058	0.814	0.420
Sleep (min)	Con (*n* = 25)	217.36 ± 77.92	229.88 ± 60.07	12.52 ± 63.42	0.987	0.334		
Deep	Exp (*n* = 26)	52.88 ± 18.59	64.38 ± 20.92	11.50 ± 28.05	2.090	0.047	2.464	0.017
Sleep (min)	Con (*n* = 25)	49.00 ± 13.75	51.68 ± 15.30	2.68 ± 18.90	0.709	0.485		

Note: Exp = experimental; Con = control; NOA = number of awakenings; PSQI = Pittsburgh Sleep Quality Index; REM = rapid eye movement; SD = standard deviation; SE = sleep efficiency; SL = sleep latency; TOA = time of awakenings; TST = total sleep time.

## Data Availability

Not applicable.

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
