# Peer review of "The Effects of Auricular Acupressure on Low Back Pain, Neuropathy and Sleep in Patients with Persistent Spinal Pain Syndrome (PSPS): A Single-Blind, Randomized Placebo-Controlled Trial"

_ijerph, 2023, doi:10.3390/ijerph20031705_

Round 1

Reviewer 1 Report

A well thought out and correctly conducted study, however I am stuck on the statistics and have the following questions for the authors:

1. the primary evaluations are done via analysis of variance with repeated measures, while the calculation of the optimal sample size, however, is done via a t-test. Since randomisation was used, the effects we are interested in would be proven via the interaction effect of the analysis of variance, right?

2. multiple testing of highly correlated variables takes place. Why was no Manova used or inflation of the alpha error counteracted with a Bonferoni correction? The current analyses leave the feeling of "fishing for significance".

The effect sizes and degrees of freedom that are generally missing everywhere would have to be added in any case.

Author Response

We are very much sincerely pleased with your comments. It seems to have become a better journal by editing according to your comments. I am very grateful. Thank you so much.

Reviewer 2 Report

Overall, this is a tight and nicely designed study. It is not really looking at FBSS, but more so, post-surgical patients. I am not sure that one can call the control group a placebo, since even if these are not acupuncture/pressure points, they could activate mechanoreceptors in the skin, so I might list that as a limitation. And I think it could be designed as double-blind, not just single-blind, but that is a minor concern. AS it stands, I think this is actually well thought out.

Author Response

(The authors gave the same response as above.)

Reviewer 3 Report

Comments on “The Effects of Auricular Acupressure on Low Back Pain, Neuropathy and Sleep in Patients with Persistent Spinal Pain Syndrome (PSPS): A Single-Blinded, Randomized Placebo-Controlled Trial”

The work done by Lim et. al. is overall well written and methodically carried out. I have a few comments that may help in further enhancing the scientific rigor of the manuscript.

1.      In the manuscript authors have used Auricular acupressure and its abbreviation AA both in different places throughout the manuscript. Please introduce the abbreviation once and use it henceforth.  Please correct.

2.      Please add references for line 61-64, as the authors state “many studies “in line 61 but however provide only the reference 15.

3.      How was 6 weeks of AA application determined to be the optimum time?

4.      Line no. 79-81 and lines 83-88, is the repetition of same information. Please use either of the lines making sure the same thing is not reiterated back-to-back.

5.      It is recommended to provide detailed information about what is VAS, DN4 and PSQ1and why are these parameters measured, in the introduction section, before introducing them in lines 99-102.

6.      Please discuss about lumbar PPT in detail.

7.       Though the authors describe PSPS, they should provide more thorough information like the statistics on the present prevalence of PSPS worldwide.

8.      Please explain why white mustard seeds were applied (line 138)

9.      What does author think about their observation on PSQ1, discuss.

10.   The overall figure 3 quality looks very poor. Please enhance the scientific rigor of the figures by consulting good quality journals that have published figures of similar kind.
